Pushing up or pushing out—an initial investigation into horizontal- versus vertical-force training on swimming start performance: a pilot study

Thng Shiqi 1 2 shiqi.thng@student.bond.edu.au
Pearson Simon 2
http://orcid.org/0000-0001-9851-1068 Keogh Justin W.L. 1 3 4 5
1 Faculty of Health Sciences and Medicine, Bond University , Gold Coast, QLD , Australia
2 Queensland Academy of Sport , Nathan, QLD , Australia
3 Sports Performance Research Centre New Zealand, Auckland University of Technology , Auckland , New Zealand
4 Cluster for Health Improvement, Faculty of Science, Health, Education and Engineering, University of the Sunshine Coast , Sippy Downs, QLD , Australia
5 Kasturba Medical College, Mangalore, Manipal Academy of Higher Education , Manipal, Karnataka , India
Doyle Tim
Electronic publication date: 2021 Feb 24
Publication date: 2021
Volume: 9
Electronic Location ID: e10937
Received 2020 Sep 21; Accepted 2021 Jan 21
Copyright: © 2021 Thng et al.
Copyright year: 2021
Copyright holder: Thng et al.
License: This is an open access article distributed under the terms of the Creative Commons Attribution License, which permits unrestricted use, distribution, reproduction and adaptation in any medium and for any purpose provided that it is properly attributed. For attribution, the original author(s), title, publication source (PeerJ) and either DOI or URL of the article must be cited.
License URL: https://creativecommons.org/licenses/by/4.0/

Keywords: Swim start, Swimming, Specificity of training, Force-vector theory, Resistance training

Funding: Bond University Faculty of Health Sciences and Medicine This work was supported by the Queensland Academy of Sport’s Sport Performance Innovation and Knowledge Excellence Unit in conjunction with Bond University Faculty of Health Sciences and Medicine. The funders had no role in study design, data collection and analysis, decision to publish, or preparation of the manuscript.

==============================
Background

The block phase in the swimming start requires a quick reaction to the starting signal and a large take-off velocity that is primarily horizontal in direction. Due to the principle of specificity of training, there is a potential benefit of performing a greater proportion of horizontal force production exercises in a swimmers’ dry-land resistance training sessions. Therefore, the purpose of this pilot study was to provide an insight into the effects of a horizontal- (HF) vs vertical-force (VF) training intervention on swim start performance.

Methods

Eleven competitive swimmers (six males (age 20.9 ± 1.8 years, body mass 77.3 ± 9.7 kg, height 1.78 ± 0.05 m) and five females (age 21.4 ± 2.0 years, body mass 67.5 ± 7.4 kg, height 1.69 ± 0.05 m)) completed 2 weekly sessions of either a horizontal- or vertical-force focused resistance training programme for 8 weeks. Squat jump force-time characteristics and swim start kinetic and kinematic parameters were collected pre- and post-intervention.

Results

Across the study duration, the swimmers completed an average of nine swimming sessions per week with an average weekly swim volume of 45.5 ± 17.7 km (HF group) and 53 ± 20.0 km (VF group), but little practice of the swim start per week (n = 9). Within-group analyses indicated a significant increase in predicted one repetition maximum (1RM) hip thrust strength in the HF group, as well as significant increases in grab resultant peak force but reductions in resultant peak force of the block phase for the VF group. No significant between-group differences in predicted 1RM hip thrust and back squat strength, squat jump force-time and swim start performance measures were observed after 8 weeks of training. Significant correlations in the change scores of five block kinetic variables to time to 5 m were observed, whereby increased block kinetic outputs were associated with a reduced time to 5 m. This may be indicative of individual responses to the different training programmes.

Discussion

The results of this current study have been unable to determine whether a horizontal- or vertical-force training programme enhances swim start performance after an 8-week training intervention. Some reasons for the lack of within and between group effects may reflect the large volume of concurrent training and the relative lack of any deliberate practice of the swim start. Larger samples and longer training duration may be required to determine whether significant differences occur between these training approaches. Such research should also look to investigate how a reduction in the concurrent training loads and/or an increase in the deliberate practice of the swim start may influence the potential changes in swim start performance.

Introduction

The important role that muscular strength and power play in enhancing swimming performance has led to the widespread adoption of dry-land resistance training modalities into a concurrent training model for competitive swimmers (Aspenes et al., 2009; Crowley, Harrison & Lyons, 2017; Haycraft & Robertson, 2015). While much of the swimming strength and conditioning research has been on the free swim portion (Crowley, Harrison & Lyons, 2017), there is now a greater focus on starts and turns since swimmers have to rapidly apply large forces on the starting block or wall to increase horizontal impulse and velocity (Born et al., 2020; Jones et al., 2018; Rebutini et al., 2014).

Changes in the starting block and starting technique may have further increased the importance of lower body strength and power for swim start performance. The OSB11 start block, which was introduced by the International Swimming Federation in 2010, has an angled kick plate at the rear of the block that enables the swimmer to adopt a kick start technique (Tor, Pease & Ball, 2015a). The additional kick plate allows for an increased duration of effective force application (i.e. greater horizontal force component) on the blocks, which can increase horizontal impulse and take-off velocity (Honda et al., 2010).

With the new OSB11 start block and kick start technique, the swim start may share some similarities to the sprint start in track and field regarding the starting position, importance of a quick reaction to the starting stimulus, and the need to produce large horizontal impulse on the starting blocks (Čoh et al., 2017; Harland & Steele, 1997). Analysis of the force-time characteristics of swimmers performing the squat jump has identified concentric impulse as a strong predictor of swim start performance as assessed by time to 5 m and 15 m (Thng et al., 2020). Further, near perfect correlations (r > 0.90) between countermovement jump height or take-off velocity and very large correlations for measures of maximal strength (r = 0.7–0.9) to swim start performance have been reported in a recent systematic review (Thng, Pearson & Keogh, 2019).

Despite the strength of this cross-sectional literature (Thng, Pearson & Keogh, 2019), there is relatively little research quantifying the chronic effects of resistance training on swim start performance. Three studies have utilised jump and plyometric exercise programmes (Bishop et al., 2009; Rebutini et al., 2014; Rejman et al., 2017), two studies (Breed & Young, 2003; Garcia-Ramos et al., 2016) used a more general resistance training programme, and one study (Born et al., 2020) compared the effects of maximal strength resistance training to plyometrics. The three plyometric studies included adolescent (Bishop et al., 2009) and national level swimmers (Rebutini et al., 2014; Rejman et al., 2017) who performed 6–9 weeks of plyometrics, twice a week. Significant improvements in time to 5 m and 5.5 m, take-off velocity, horizontal forces and impulse were observed as a result of these plyometric exercise programmes (Bishop et al., 2009; Rebutini et al., 2014; Rejman et al., 2017). In contrast, the remainder of these plyometric and resistance training studies typically reported no significant changes in time to 5 m or 15 m, or any block phase kinetic or kinematic characteristics (Born et al., 2020; Breed & Young, 2003; Garcia-Ramos et al., 2016). The only exception to this was the significant improvements in time to 5 m and 15 m observed for the subset of under 17-year-old swimmers who performed maximal strength training, with no such effects reported for the under 17-year-old plyometric group (Born et al., 2020).

A possible explanation for the uncertainty regarding whether jump/plyometric or more general resistance training programmes produces greater improvements in swim start performance may reflect the direction-specific nature of resistance training. In a review by Randell et al. (2010) on the specificity of resistance training to sports performance, it was proposed training adaptations may be direction-specific, and that athletes who are required to apply forces in the horizontal plane should perform several exercises containing a horizontal component. More recently, this directional specificity of training has been referred to as the force-vector theory (Fitzpatrick, Cimadoro & Cleather, 2019), with the hip thrust and prowler push/heavy sled pull being two of the most commonly used horizontal-force exercises (Contreras et al., 2017; Fitzpatrick, Cimadoro & Cleather, 2019; Morin et al., 2017; Winwood et al., 2015). A study by Contreras et al. (2017) using the hip thrust significantly improved 10 m and 20 m sprint running times (−1.05% and −1.67%, respectively) compared to the front squat, which is a vertical-force exercise (+0.10% and −0.66%, respectively). The prowler push, which requires the athlete to push a loaded sled in the horizontal plane, has been shown to closely mimic the horizontal plane power requirements of sprinting (Tano et al., 2016). A study involving 30 sub-elite rugby players observed that a horizontal-focused resistance training programme including the prowler push significantly improved performance in a number of strength, sprinting, and change of direction tests (Winwood et al., 2015). However, no significant between-group effects were observed between the horizontal-focused and traditional resistance training programmes (Winwood et al., 2015).

The potential direction specificity of resistance training exercises for improving aspects of swim start performance has been examined in two jump and plyometric training studies (Rebutini et al., 2014; Rejman et al., 2017) and two acute training studies utilising post-activation potentiation (PAP) (Cuenca-Fernandez, Lopez-Contreras & Arellano, 2015; Cuenca-Fernández et al., 2018). Rebutini et al. (2014) and Rejman et al. (2017) observed a 10.4% and 13.8% increase in take-off velocity in the swim start post 9- and 6-weeks of plyometric training, respectively, that included a variety of horizontal jumps. Acute improvements in time to 5 m (Cuenca-Fernandez, Lopez-Contreras & Arellano, 2015; Cuenca-Fernández et al., 2018) and 15 m (Cuenca-Fernandez, Lopez-Contreras & Arellano, 2015) after performing PAP protocols that were biomechanically similar to the foot position in the kick start on the OSB11 start block have also been observed. However, out of these four plyometric and PAP studies, only one (Cuenca-Fernandez, Lopez-Contreras & Arellano, 2015) utilised the OSB11 start block and the kick start technique currently used by high performance swimmers.

Therefore, the primary aim of this pilot study was to gain some preliminary insight into the comparative effects of a horizontal- vs vertical-force resistance training programme on swim start performance and squat jump (SJ) force-time characteristics. A secondary aim of the study was to better understand how changes in certain SJ force-time characteristics may be correlated with the changes in swim start performance in competitive swimmers.

Materials and Methods

Experimental design

An 8-week training programme sought to examine how a horizontal-force (HF) compared to vertical-force (VF) oriented emphasis resistance training programme would potentially alter swim start performance. Participants were randomly assigned to either a HF or VF training group (HF: n = 6, VF: n = 7), with each group performing two resistance training sessions per week.

Participants

Thirteen participants (8 males (age 21.0 ± 1.6 years, body mass 78.6 ± 8.3 kg, height 1.80 ± 0.06 m), and 5 females (age 21.4 ± 2.0 years, body mass 67.5 ± 7.4 kg, height 1.69 ± 0.05 m)) volunteered to participate in this study. Participants were national level swimmers with at least 4 years’ experience in competing in national championships and at least 1 year of land-based resistance training experience that included the barbell back squat and hip thrust under the supervision of a strength and conditioning coach. Participants with any known contraindication to maximal training performance and/or injuries that would interfere with their ability to complete the study or compromise their health and wellness were excluded. Prior to participating in this study, participants were briefed on the experimental design and gave written informed consent to participate in the study. This investigation was conducted in accordance with the Declaration of Helsinki and approved by Bond University Human Research Ethics Committee (00088).

Assessments were conducted at baseline (week one) and the end of the training programme (week nine). Participants were instructed to maintain their nutritional and sleep habits, and to avoid alcohol and caffeine consumption for at least 24 h before testing sessions. All tests were performed on the same day of the week between 7:00 am and 11:00 am. Participants reported to the gymnasium to perform the squat jump test prior to the swim start performance test.

Training intervention

The training programme was organised into two phases. In the first phase (weeks one to four), each group performed three HF and VF lower body exercises, respectively. A direction specific lower body jump was added in the second phase for each group (weeks five to eight) (Table 1). The HF training group was prescribed a ‘start jump’ which is a jump for horizontal distance initiated from a mimicked swim start position (Fig. 1), while the VF training group performed the squat jump. When performing the jumps, the HF group were instructed to jump as far forward as possible, while the VF group were instructed to jump as high as possible with each jump.

Table 1 An outline of the 8-week intervention programme for the Horizontal-Force (HF; n = 6) and Vertical-Force (VF; n = 5) training group with weekly sets, repetition, and load progression for the lower body strength and jumping exercises.

Intervention Group	Day	Exercise	Training focus	
			Strength	Strength-power				
			Training week	
			1	2	3	4	5	6	7	8	
			Sets × reps	Sets × reps	Sets × reps	Sets × reps	Sets × reps	Sets × reps	Sets × reps	Sets × reps	
HF group	1a	Barbell hip thrust	3 × 8	3 × 8	3 × 6	2 × 6	3 × 5	3 × 5	3 × 4	2 × 4	
	1b	‘Start’ jump					3 × 3	3 × 3	3 × 3	2 × 3	
	2a	Prowler push^	3 × 8	3 × 8	3 × 6	2 × 6	3 × 5	3 × 5	3 × 4	2 × 4	
	2b	Drop vertical jump					3 × 3	3 × 3	3 × 3	2 × 3	
VF group	1a	Back squat	3 × 8	3 × 8	3 × 6	2 × 6	3 × 5	3 × 5	3 × 4	2 × 4	
	1b	Squat jump					3 × 3	3 × 3	3 × 3	2 × 3	
	2a	Rear foot elevated split squat^	3 × 8	3 × 8	3 × 6	2 × 6	3 × 5	3 × 5	3 × 4	2 × 4	
	2b	Drop vertical jump					3 × 3	3 × 3	3 × 3	2 × 3	
Note:

^ Repetitions listed are for each leg.

Figure 1 Initial positioning of the ‘start’ jump for the Horizontal-Force (HF) training group.

Participants performed the training programme utilising sets and repetition ranges typically used for developing maximal strength (Bird, Tarpenning & Marino, 2005). Participants followed two 4-week mesocycles using a 3:1 loading paradigm, with a progressive increase in load for the first 3 weeks followed by a reduction in load in the fourth week (Turner, 2011). This was considered important as the swimmers were still maintaining high volumes of swimming training throughout the intervention. As the majority of propulsive forces in the free swim phase comes from the upper body (Morouço et al., 2015), both groups also performed three sets of several upper body exercises including pull-ups, bench pull or seated row; and three sets of exercises for the abdominals/lower back region, as successfully used by Contreras et al. (2017) in a previous horizontal- vs vertical-force direction study. Sets were separated by a 1-min rest period (Ritchie et al., 2020). Training records were kept for each participant to analyse the load progression of the training programme. Predicted one repetition maximum (1RM) of the hip thrust and barbell back squat was calculated pre- and post-intervention using the Brzycki equation: Predicted 1RM = weight lifted/1.0278−0.0278 (no. of repetitions) (Brzycki, 1993). Repetition ranges used in the predicted 1RM was performed during the first training session (estimated from eight repetitions) and at the last training session (estimated from four repetitions). Participants were asked to refrain from performing any additional resistance training and to maintain their current diet for the course of this study.

Squat jump test

The SJ test was collected as previously described by Thng et al. (2020). All participants completed a standardised dynamic warm-up consisting of a predetermined series of dynamic joint ranges of motion of the upper and lower body under the supervision of a strength and conditioning coach. Participants were then given two practice SJs before the test was conducted. All SJs were performed on a force platform (FD4000; ForceDecks, London, United Kingdom), with a sample rate of 1,000 Hz. Participants started in an upright standing position with their hands on their hips and were instructed to keep their hands on their hips to prevent the influence of any arm movements for the jump trials. All participants were instructed to adopt a squat position using a self-selected depth that was held for 3 seconds before attempting to jump as high as possible (Mitchell et al., 2017). A successful trial was one that did not display any small amplitude countermovement at the start of the jump phase on the force trace (Sheppard & Doyle, 2008). All participants performed three maximal effort SJs with a 30-s passive rest between each effort. The SJ trial with the highest jump height was kept for data analysis. Jump height was determined by the flight-time method (Jump height = g*t2/8, where g is the acceleration due to gravity and t is the flight time) (Linthorne, 2001). Ground reaction force data from the SJs were analysed using the commercially available ForceDecks software (ForceDecks, London, United Kingdom). A description of the SJ variables that were identified by Thng et al. (2020) as significant predictors of swim start performance were extracted for analysis are provided in Table 2.

Table 2 Description of squat jump variables obtained from the ForceDecks force platform, and the swim start variables obtained from the KiSwim Performance Analysis System.

	Variable	Description	
ForceDecks SJ variables	Concentric impulse (N.s.)	Net impulse of vertical force during the concentric phase	
Concentric mean power (W)	Mean power during concentric phase	
Concentric rate of power development (RPD) (W/s)	Rate of power development between start of concentric phase to peak power	
Jump height (cm)	Jump height calculated from Flight Time (time between take-off and landing) in centimetres	
Reactive strength index modified (RSImod) (m/s)	Jump height (Flight Time) divided by contraction time	
KiSwim swim start kinetic variables	Average acceleration (m/s/s)	Horizontal take-off velocity/seconds from starting gun to take-off	
Average power (W/kg)	The average power relative to the swimmers’ body mass produced from the starting signal to when the swimmer leaves the starting block. This was calculated as the product of (absolute force × absolute velocity)/body mass	
Horizontal take-off velocity (m/s)	The horizontal take-off velocity calculated by integrating horizontal acceleration	
Work/kg (J/kg)	Average power × seconds from the starting gun to take-off	
Front horizontal peak force (N)	Peak horizontal force on the front plate of the starting block (grab bar component not subtracted)	
Grab resultant peak force (N/BW)	Peak grab bar resultant force	
Rear horizontal peak force (N)	Peak horizontal force on the foot plate (grab bar component not subtracted)	
Total resultant peak force (N)	Peak resultant force (grab bar component subtracted)	
Rear resultant average force (N/BW)	Average resultant force on the foot plate (grab bar component not subtracted)	
Swim start performance times	Time to 5 m and 15 m (s)	Time from the starting signal to a swimmers’ head crossing the 5 m and 15 m mark. This is digitised at the point where the centre of the swimmers’ head crosses 5 m and 15 m	

Swim start performance test

Swim starts were collected using methods as described by Thng et al. (2020). Prior to the swim start test, all swimmers completed a pool-based warm-up based on their usual pre-race warm-up routine. Participants then performed three maximal effort swim starts to 15 m with their main swim stroke (front crawl (n = 8), butterfly (n = 3), or breaststroke (n = 2)) and preferred kick plate position, which was recorded to ensure consistency between testing sessions. Trials were started as per competition conditions and swimmers were instructed to swim to a distance past the 15 m mark, in order to ensure that representative values at the 15 m distance were obtained (Barlow et al., 2014). Two-minutes of passive recovery were given between each trial (Tor, Pease & Ball, 2015b). The start with the fastest 15 m time was selected for further analysis. Swim starts were collected using a Kistler Performance Analysis System—Swimming (KiSwim, Kistler Winterthur, Switzerland), which utilises a force instrumented starting block, constructed to match the dimensions of the Omega OSB11 block (KiSwim Type 9691A1; Kistler Winterthur, Switzerland). Time to 5 m and 15 m were collected using five calibrated high speed digital cameras operating at 100 frames per second, synchronised to the instrumented KiSwim starting block. One camera was positioned 0.95 m above the water and 2.5 m perpendicular to the direction of travel to capture the start and entry of swimmer into the water, while the other three cameras were positioned 1.3 m underwater at 5 m, 10 m and 15 m perpendicular to the swimmer to capture the time to 15 m. The times to 5 m and 15 m were defined as the time elapsed from the starting signal until the apex of the swimmer’s head passed the respective distances (Tor, Pease & Ball, 2015b). An Infinity Start System (Colorado Time Systems, Loveland, CO, USA) provided an audible starting signal to the athletes and an electronic start trigger to the KiSwim system. Kinetic and kinematic variables of block performance extracted for analysis were identified by Thng et al. (2020) as key predictors of time to 5 m and 15 m (Thng et al., 2021, unpublished data). A description of the swim start variables analysed are provided in Table 2.

Statistical analysis

Descriptive statistics are reported as mean ± SD for normally distributed continuous variables and frequencies for categorical variables. Normality was checked using histograms, normal Q–Q plots, and the Shapiro–Wilk test. A paired sample t-test was used to determine whether statistically significant differences were found between pre- and post-test means within each group. Independent t-tests were carried out to test for the difference in change in the outcome between intervention groups. Effect sizes (ES) with 95% confidence intervals (95% CI) were reported in standardised (Cohen’s d) units as the change in mean to quantify the magnitude of differences within (i.e. post-intervention—pre-intervention results) and between the two intervention groups (i.e. HF and VF). Criteria to assess the magnitude of observed changes were: 0.0–0.2 trivial; 0.20–0.60 small; 0.60–1.20 moderate; and >1.20 large (Hopkins, 2002). Effect sizes were calculated using a programme created by Lenhard & Lenhard (2016).

To gain some preliminary insight into how changes in the SJ force-time characteristics may be correlated with the changes in swim start performance, the association between the change scores (calculated as the difference between each individuals’ pre- and post-test scores) for these outcomes were assessed by Pearson’s product-moment correlation coefficient (r). Data were analysed with SPSS version 23.0.0 (SPSS Inc., Chicago, IL, USA). P-values < 0.05 were deemed to indicate statistical significance.

Results

Training compliance

Of the 13 initial participants, 11 participants completed the training study (Table 3). Two participants were removed due to moving to another swim squad (n = 1) and non-adherence to the training protocol (n = 1). Participants completed a total of 14 ± 3 out of 16 training sessions, with the primary reasons for missed training sessions being short-term illness or domestic competitions. A summary of the within-group and between-group changes are provided in Table 4.

Table 3 Physical characteristics of participants (N = 11).

Variables	HF group (n = 6)	VF group (n = 5)	
Age (years)	21.3 ± 1.7	21.0 ± 2.2	
Sex (male/female)	3/3	3/2	
Body mass (kg)	74.3 ± 10.5	70.0 ± 10.3	
Height (m)	1.73 ± 0.06	1.74 ± 0.08	
Weekly in-water training volume (km)	45.5 ± 17.7	53.0 ± 20.0	
Weekly number of swim starts performed	9 ± 2	9 ± 2	
Note:

All data, apart from the sex of the participants are presented as means and standard deviations.

Table 4 Pre- (week 1) and post- (week 9) measures of squat jump force-time variables and swim start kinetic and kinematic parameters for the horizontal-force (HF) and vertical-force (VF) training groups.

Results are presented as mean ± SD except for effect sizes and change scores.

	HF group (n = 6)	VF group (n = 5)	Between-group differences	
	Week 1	Week 9	Change scores	Within-group ES (95% CI)	Week 1	Week 9	Change scores	Within-group ES (95% CI)	Mean difference (95% CI)	ES (95% CI)	
Predicted 1RM strength	
Hip thrust (kg)	78.5 ± 15.0	118.3 ± 26.9	39.8 ± 16.6**	1.83 [−0.08 to 3.73]							
Barbell back squat (kg)					70.6 ± 27.0	85.20 ± 38.67	14.6 ± 20.8	0.44 [−1.34 to 2.21]	25.23 [−0.23 to 50.70]	1.36 [0.04–2.67]	
SJ force-time variables	
Jump height (cm)	28.4 ± 7.5	29.1 ± 7.0	0.8 ± 3.1	0.11 [−1.50 to 1.71]	29.0 ± 10.7	27.1 ± 8.3	−1.9 ± 2.9	−0.19 [−1.95 to 1.56]	2.63 [−1.50 to 6.76]	0.87 [−0.37 to 2.11]	
Concentric impulse (N.s.)	183.2 ± 46.2	182.3 ± 49.4	−0.9 ± 7.6	−0.02 [−1.62 to 1.58]	167.3 ± 43.3	165.3 ± 44.1	−2.0 ± 8.4	−0.05 [−1.80 to 1.71]	1.06 [−9.84 to 11.97]	0.14 [−1.05 to 1.33]	
RSImod (m/s)	0.79 ± 0.16	0.73 ± 0.21	−0.07 ± 0.10	−0.32 [−1.93 to 1.29]	0.75 ± 0.30	0.73 ± 0.33	−0.02 ± 0.14	−0.06 [−1.82 to 1.69]	−0.04 [−0.20 to 0.12]	−0.42 [−1.62 to 0.78]	
Concentric mean power (W)	1414.2 ± 387.6	1442.0 ± 527.8	27.8 ± 174.6	0.06 [−1.54 to 1.66]	1268.0 ± 437.5	1241.0 ± 587.7	−27.0 ± 254.8	−0.05 [−1.81 to 1.70]	54.8 [−238.3 to 347.9]	0.26 [−0.94 to 1.45]	
Concentric RPD (W/s)	11986.3 ± 2879.3	10130.6 ± 3817.3	−1,855.6 ± 1921.3	−0.55 [−2.18 to 1.08]	10216.0 ± 5333.5	10874.5 ± 6109.3	658.4 ± 3017.4	0.12 [−1.64 to 1.87]	−2,514.1 [−5896.6 to 868.3]	−1.02 [−2.28 to 0.24]	
KiSwim kinetic variables	
Average Power (W/kg)	19.66 ± 3.33	19.52 ± 2.94	−0.15 ± 0.63	−0.05 [−1.65 to 1.56]	20.65 ± 5.42	19.91 ± 5.05	−0.74 ± 0.97	−0.14 [−1.90 to 1.61]	0.59 [−0.50 to 1.68]	0.74 [−0.49 to 1.97]	
Average Acceleration (m/s/s)	6.20 ± 0.80	6.15 ± 0.64	−0.04 ± 0.22	−0.07 [−1.67 to 1.53]	6.42 ± 1.14	6.26 ± 1.04	−0.16 ± 0.26	−0.15 [−1.90 to 1.61]	0.12 [−0.21 to 0.45]	0.50 [−0.70 to 1.71]	
Work/kg (joules)	13.83 ± 2.00	13.91 ± 1.93	0.08 ± 0.43	0.04 [−1.56 to 1.64]	13.73 ± 2.68	13.57 ± 2.51	−0.16 ± 0.39	−0.06 [−1.82 to 1.69]	0.24 [−0.32 to 0.80]	0.58 [−0.63 to 1.79]	
Horizontal take-off velocity (m/s)	4.36 ± 0.38	4.38 ± 0.36	0.03 ± 0.14	0.05 [−1.55 to 1.66]	4.29 ± 0.46	4.29 ± 0.41	0.00 ± 0.09	0.00 [−1.75 to 1.75]	0.03 [−0.13 to 0.19]	0.25 [−0.94 to −1.44]	
Total resultant peak force (N/BW)	1.73 ± 0.21	1.68 ± 0.19	−0.05 ± 0.07	−0.25 [−1.86 to 1.36]	1.95 ± 0.53	1.84 ± 0.55	−0.11 ± 0.06*	−0.20 [−1.96 to 1.55]	−0.06 [−0.15 to 0.03]	0.91 [−0.33 to 2.16]	
Front horizontal peak force (N/BW)	0.69 ± 0.07	0.70 ± 0.05	0.02 ± 0.05	0.16 [−1.44 to 1.77]	0.73 ± 0.05	0.72 ± 0.09	−0.01 ± 0.05	−0.14 [−1.89 to 1.62]	−0.03 [−0.09 to 0.04]	0.60 [−0.61 to 1.81]	
Rear horizontal peak force (N/BW)	0.90 ± 0.19	0.88 ± 0.16	−0.02 ± 0.05	−0.11 [−1.72 to 1.49]	0.91 ± 0.16	0.92 ± 0.15	0.01 ± 0.05	0.06 [−1.69 to 1.82]	0.03 [−0.03 to 0.10]	−0.60 [−1.81 to 0.61]	
Rear resultant average force (N/BW)	0.58 ± 0.10	0.58 ± 0.09	−0.01 ± 0.03	0.00 [−1.60 to 1.60]	0.58 ± 0.13	0.57 ± 0.13	−0.01 ± 0.03	−0.08 [−1.83 to 1.68]	0.00 [−0.04 to 0.04]	0.00 [−1.19 to 1.19]	
Grab resultant peak force (N/BW)	38.67 ± 7.76	38.83 ± 7.65	0.17 ± 4.17	0.02 [−1.58 to 1.62]	36.20 ± 7.92	38.80 ± 8.26	2.60 ± 1.14**	0.32 [−1.44 to 2.09]	2.43 [−1.95 to 6.81]	−0.76 [−1.99 to 0.47]	
Swim start performance times	
T5 m (s)	1.60 ± 0.15	1.61 ± 0.14	0.02 ± 0.03	0.07 [−1.53 to 1.67]	1.59 ± 0.19	1.61 ± 0.19	0.02 ± 0.03	0.11 [−1.65 to 1.86]	0.00 [−0.04 to 0.04]	0.00 [−1.19 to 1.19]	
T15 m (s)	7.33 ± 0.69	7.32 ± 0.57	−0.01 ± 0.19	−0.02 [−1.62 to 1.59]	6.82 ± 0.91	6.85 ± 0.88	0.04 ± 0.08	0.03 [−1.72 to 1.79]	−0.04 [−0.28 to 0.19]	−0.33 [−1.53 to 0.86]	
Notes:

* p < 0.05.

** p < 0.01.

BW, bodyweight; 95% CI, confidence interval of the differences within and between measures; ES, effect size; RPD, rate of power development; SD, standard deviation; SJ, squat jump.

For within group effects, a positive change score and effect size indicated that the post test score was larger than the pre-test score. For between group effects, a positive effect size indicated that the HF group had a larger change than the VF group.

Bolded values indicate an effect size difference of moderate or large.

Within-group changes post-intervention

Only three significant within-group differences were observed across both groups. For the HF group, a significant increase in predicted 1RM hip thrust strength (p = 0.04) was observed. The VF group had a significant increase in KiSwim grab resultant peak force (p = 0.007) and a significant decrease in KiSwim resultant peak force (p = 0.02).

Between-group changes post-intervention

A greater increase in predicted 1RM strength for the hip thrust was observed in the HF training group (50%) than the increase in back squat strength for the VF training group (18%) after 8 weeks of training (ES = 1.36). Moderate effect sizes were observed in two SJ force-time variables and five KiSwim variables (Table 4). Specifically, moderate effect size improvements in SJ jump height and three swim start kinetic measures were observed in the HF group. In the VF group, SJ concentric RPD and two swim start kinetic measures favoured moderate effect size improvements in the VF group.

When looking at individual changes across both groups, no significant correlations were observed between the change scores in any of the ForceDecks outcome measures and time to 5 m or 15 m. Similarly, there were no significant correlations in the change score correlations between the KiSwim outcomes and time to 15 m. However, significant correlations between the change scores for five KiSwim outcomes and time to 5 m were observed. These were average acceleration (r = −0.82, p = 0.02), horizontal take-off velocity (r = −0.81, p = 0.03), average power (r = −0.77, p = 0.05), work (r = −0.74, p = 0.01) and rear resultant average force (r = −0.71, p = 0.02).

Discussion

The present pilot study was designed to provide some insight into the potential directional specificity of resistance training (now referred to as the force-vector theory) on swim start performance and squat jump (SJ) force-time characteristics in competitive swimmers. This was achieved by examining the within- and between-group training-related changes in swim start performance for two groups of competitive swimmers, who differed on whether they performed a horizontal- or vertical-force oriented emphasis resistance training programme.

Relatively few significant within-group changes in any outcome measures were observed, with the non-significant changes being trivial to small in their effect sizes. The three significant within-group changes included significant increases in predicted 1RM hip thrust strength for the HF group as well as significant increases in swim start grab resultant peak force but reductions in resultant peak force for the VF group. No significant between-group differences were observed between the HF and VF groups in predicted 1RM strength, SJ force-time and swim start performance measures post-intervention. However, seven moderate between-group effect size differences were observed, with four outcome measures favouring greater improvements for the HF group and three outcome measures favouring the VF group. As such, this current study has been unable to determine whether the inclusion of horizontally oriented exercises has any clear benefit to swim start performance over more conventional vertically oriented exercises.

Possible explanations for our lack of significant within- or between-group improvements may include the small number of participants and short duration of the training intervention, inclusion of plyometric and non-plyometric jumps in only the last four of 8 weeks of training, the interference effect due to concurrent training and the relative complexity of the swim start. Regarding the length of the intervention, the absence of any significant improvements in swim start performance in the current study was consistent with some studies involving 21 (Born et al., 2020) or 23 (Breed & Young, 2003) participants performing 6–8 weeks of resistance training, but inconsistent with other plyometric training studies of 6–9 weeks involving nine (Rejman et al., 2017), 10 (Rebutini et al., 2014) or 22 (Bishop et al., 2009) participants.

The potentially greater adaptations in swim start performance observed in previous plyometric studies may reflect the between study differences in plyometrics training volume. The present study only included 33 jumps, compared to previous successful plyometric studies (Bishop et al., 2009; Rebutini et al., 2014; Rejman et al., 2017), which included ~484–883 jumps across the study. Interestingly, even though Born et al. (2020) included comparable volumes of plyometrics in their training study (~360–588 jumps) to those of the successful studies, the plyometric training group reported no significant improvements in swim start performance. While it cannot be discounted that the present study included an insufficient volume of plyometric exercise, the lack of any widespread changes in lower body force-time characteristics and swim start performance metrics observed in the present study and some of the literature (Born et al., 2020; Breed & Young, 2003), may be indicative of the challenges coaches face in making any substantial improvements in strength and power characteristics that transfer to improved sporting performance within such short periods of concurrent training.

Concurrent training is complex in that both swim training and resistance training impose different acute stresses on the body that elicit distinct adaptations. In particular, the concurrent development of both muscular strength/power and aerobic endurance from resistance training and swimming training respectively can lead to conflicting neuromuscular adaptations (Garcia-Pallares et al., 2009). In the current study, participants were primarily middle to long distance swimmers, who performed nine in-water sessions weekly (HF: 45.5 ± 17.7 km and VF: 53 ± 20.0 km per week). The sessions had an average swimming volume of 5.1 km and 5.8 km for the HF and VF group per session, with two swimming sessions a day performed several days per week. In contrast, the resistance training programme was only performed twice per week. The interference effect from concurrent training is more likely observed with ≥ three sessions of high volume endurance training weekly (Bishop et al., 2019). Therefore, the high aerobic training volume for the participants in the present study likely attenuated any resistance training-induced adaptations. Consistent with this view, Haycraft & Robertson (2015) recommend swim training volumes be reduced ≤5 km per day to enable maximal strength and power gains and minimise neuromuscular fatigue.

It should also be acknowledged that the swim start is a discrete skill, requiring a quick reaction to the starting stimulus and the ability to effectively coordinate hand and foot forces to optimise horizontal impulse and take-off velocity. Unfortunately, the swimmers in the present study only performed a small number of swim starts per week (n = 9 ± 2), with this performed either during regular swim training or at the end of the session. It was also interesting to observe that Born et al. (2020) also reported a low volume of swim starts (n = 16) performed per week. Breed & Young (2003) emphasised that a higher skill component is involved in executing the swim start in comparison to vertical jump. This may reflect the requirement for how the ankle, knee, and hip joint moments needs to be coordinated effectively with those of the upper body during the block phase to maximise horizontal take-off velocity. Further, minimising the time to 15 m also requires a clean entry into the water and a streamlined glide position with undulatory leg kicks to minimise velocity loss while transitioning into the break-out of full swimming and stroking after 15 m (Vantorre, Chollet & Seifert, 2014). The relative absence of deliberate practice of the swim start coupled with performing the starts in a fatigued state may also help explain the minimal transfer of the resistance training interventions to improved swim start performance in the current study and that of Born et al. (2020). However, significant correlations in the change scores of five block kinetic variables to time to 5 m were observed in the current study, whereby an increase in block kinetic variables was associated with a decrease in time to 5 m. Such correlations suggest that the longitudinal tracking of individual swimmers’ SJ force-time characteristics may provide some insight into their potential improvements in swim start performance.

Due to the demands of competitive swimming, it seems necessary that a targeted approach of both resistance training and deliberate practice of the swim start is required across the annual periodisation plan to improve swim start performance. This is especially important to minimise the potential adverse effects of concurrent training and maximise skill acquisition, particularly for swimmers who need to improve aspects of their swim start technique, given the complexity of the swim start. Practical recommendations include a targeted block of resistance training focused on improving the strength and power characteristics required for the swim start in a low swimming volume phase such as pre-season for a longer duration than used in the present study. Specifically, extended intervention periods >6 months have been suggested for an optimal transfer of strength and power qualities to performance in well-trained endurance athletes (Beattie et al., 2014). Incorporating greater amounts of deliberate practice of swim starts, especially at the beginning of each training session when the swimmer is mentally and physically fresh would appear to be beneficial for skill acquisition (Branscheidt et al., 2019).

Conclusion

There were very few significant differences observed, either within or between the HF and VF groups after an 8-week training intervention on swim start performance. Despite exploring the inclusion of a higher proportion of horizontally oriented exercises based on the force-vector theory, the current study did not observe a transfer to improved swim start performance. However, this should not discount the potential value of including horizontally directed exercises to improve swim start performance, given the results were similar to those from more traditional vertically oriented exercises. Future studies should consider an extended training intervention completed during a phase of lower swim training volume to enable strength and power adaptions to occur.

Supplemental Information

Supplemental Information 1 Raw data of squat jump force-time variables and swim start kinetic and kinematic variables.

Click here for additional data file.

This work was supported by the Queensland Academy of Sport’s Sport Performance Innovation and Knowledge Excellence Unit in conjunction with Bond University Faculty of Health Sciences and Medicine. The authors would like to acknowledge Mr. Andrew Pyke for his assistance with data collection and coach Mr. Adam Mallet for allowing his athletes to be a part of this study. The authors also wish to thank Ms. Evelyne Rathbone for her statistical assistance in this study and resulting manuscript. There is no conflict of interest related to the content of this article.

Additional Information and Declarations

Competing Interests

Author Contributions

Human Ethics

Data Availability

Justin W.L. Keogh is an Academic Editor for PeerJ.

Shiqi Thng conceived and designed the experiments, performed the experiments, analysed the data, prepared figures and/or tables, authored or reviewed drafts of the paper, and approved the final draft.

Simon Pearson conceived and designed the experiments, performed the experiments, analysed the data, authored or reviewed drafts of the paper, and approved the final draft.

Justin W.L. Keogh conceived and designed the experiments, analysed the data, authored or reviewed drafts of the paper, and approved the final draft.

The following information was supplied relating to ethical approvals (i.e. approving body and any reference numbers):

Bond University Human Research Ethics Committee approved this research (00088).

The following information was supplied regarding data availability:

Raw data, including pre- and post-intervention measures of squat jump force-time measures and kinetic and kinematic variables of the swim start, are available as a Supplemental File.

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
