# Peer review of "Pushing up or pushing out—an initial investigation into horizontal- versus vertical-force training on swimming start performance: a pilot study"

_PeerJ, doi:10.7717/peerj.10937_

## Round 0.1 · original submission · Major Revisions

· Academic Editor

Major Revisions

Thank you for submitting your research to PeerJ. It has now been reviewed and it has been suggested by both reviewers that there remains some work to be done to make it suitable for publication. Please refer to their comments and I look forward to receiving a revised manuscript.

·

Basic reporting

With the exceptions of some minor points that I provide in the 'general comments' section, I thought that the authors did a really good job with their reporting of this study. The reporting is neat, clear and concise. Typically the language and style is appropriate and provides a professional feel to the manuscript.

Experimental design

In many respects this is a very well designed study. There appears to be a clear flow from the development of the research question to the design of the study. The one issue I do have though however is the reference to using 'plyometric' exercise in the intervention but then prescribing a non-plyometric exercise. It is my understanding that a plyometric exercise requires use of the stretch-shortening cycle and, unless I've missed something (and that's very possible) the exercises prescribed in this study's intervention do not.

Validity of the findings

I think that the authors do a good job with their findings. However, the point I raised in the previous section regarding plyometric exercise needs to be addressed to fully satisfy this part of my review.

Additional comments

Thank you for giving me the opportunity to review your manuscript. Below I outline some specific points that I would like the authors to address and I refer to the line numbers throughout:

Line 40: please replace 'having' with 'have'

Line 45: please replace 'allows the' with 'enables' as the former suggests that permission is required

Line 51: consider replacing 'high levels of horizontal impulse' with something like 'large horizontal impulse'

Line 74: 'equivalence' could be a little misleading here... please consider replacing with a clearer word

Line 110: no doubt pedantic, but I don't think that 'comprising' is the right word here. Perhaps replace with something like '...an eight week training study...', removing reference to 'pilot' as this is more than that

Line 134-141: this is where I get a little confused. Of course, this might be due to poor understanding on my part, but I struggle to see how 'squat jump' can be plyometric given this requires use of the stretch-shortening cycle. If I'm wrong please revise this part for clarity (regarding 'squat jump' technique) otherwise delete reference to plyometric throughout and revise the manuscript accordingly

Line 179-180: I appreciate that the word count can quite often be tight, but I think it could be useful to outline these variables rather than referring the reader to another publication; perhaps present these (and brief definitions) in a table? This applies to line 201-204 too.

Line 209: what was the rationale for using paired sample t tests? Wouldn't it have been more statistically robust to use a repeated measures anova?

Line 218-19: please replace 'characteristics may correlated' with 'characteristics may be correlated'

Line 246-47: perhaps refer to the relevant figure/table here for clarify?

Line 265-66: I'm afraid that I have a problem with this wording. Perhaps largely semantics, but I think it might be more appropriate to replace these with 'horizontal' or 'vertical force orientated emphasis' because large vertical forces will still be required in the horizontal tasks

Line 280: did you really include 'plyometrics'? Please clarify and revise accordingly

Line 313: please replace 'allow for' with 'enable'; same on line 358

·

Basic reporting

The article is well-written and uses clear language.

The introduction provides sufficient background to demonstrate how the work fits into the broader field of knowledge and references relevant prior literature.

The article structure is acceptable and conforms to PeerJ’s suggested format. The figure is relevant to the content of the article. The figure could potentially be improved by including pictures of all 8 exercises, but only if the authors and editors suspect PeerJ readers might be unfamiliar with the exercises.

In Table 1, was the drop jump in the HF group a drop *broad* jump, and the drop jump in the VF group a drop *vertical* jump?

For Table 3, it would be helpful to specify that the force-time characteristics were for a squat jump.

Raw data were made available, although it looks like some demographics variables (e.g. age) and the hip thrust and back squat strength data are missing. I suggest including a “codebook” with the raw data, perhaps in a separate tab, where the variable names are at least defined if not also briefly explained.

The submission is self-contained. However, it would be helpful if the equation from Brzycki 1993 for calculating estimated 1RM were provided so the reader doesn’t have to chase down that reference.

Please check the formatting for your references. For the article titles, be sure to adhere to PeerJ’s formatting instructions for title case vs. just first letter capitalized.

Experimental design

This original primary research is within the Aims and Scope of PeerJ, and the research appears to have been conducted ethically and rigorously.

The research question is well-defined, relevant, and meaningful, and the authors state how the research fills a gap. However, there is a secondary aim that appears somewhat by surprise in the statistical analysis section ("To gain some preliminary insight into how changes in the SJ force-time characteristics may correlated with the changes in swim start performance”). I suggest stating this as a secondary aim at the end of the introduction (and fixing the typo therein). Alternatively, consider removing all aspects of this analysis from the manuscript if it does not rise to the level of importance to be considered a secondary aim.

The methods were generally described with good detail, although there were several areas for improvement:

1. The description of the participants in lines 112 and 123 are duplicitous. I recommend removing the participant information from the experimental design section (line 112).

2. The methods would be more cohesive if written chronologically (e.g., moving lines 116-120 to later in the Methods section).

3. In the Participant section, were there any exclusion criteria? Also, what was the participants’ previous experience with hip thrusts and back squats?

4. Please provide rest times between sets in the training intervention (if they were prescribed) (Table 1) as well as the repetition range used in the strength testing that was then input to estimate 1RMs (lines 155-157).

5. For the SJs (lines 174-175) and swim starts (line 190), why did you take the best trials and not the average of the three trials?

6. In lines 179 and 203, it would be helpful to at least know how many SJ and block performance variables were assessed (if not indicating what the variables were, perhaps in a table). That way when the reader gets to the results, they can contextualize the 7 comparisons for which there were moderate effect sizes.

Validity of the findings

The data appear robust. However, considering these are national level athletes, I wonder about the unimpressive squat jump performances, ranging from only 18.6 cm to 43.8 cm. I recognize it’s a non-countermovement jump and these are swimmers (not necessarily the best athletes on dry land), but the values still seem low.

In terms of the statistical analysis, I question the use of inferential statistics for a pilot study with such a small sample size. Lack of statistically significant findings is not surprising for such a small sample. In general, in the results and discussion, I recommend placing greater emphasis on the effect sizes rather than the lack of statistical significance.

Also in terms of the statistics, if you choose to use null hypothesis significance testing with an a priori significance level, it’s best not to interpret p-values close to 0.05 as “trends” towards significance (line 243).

At points, the discussion and conclusion get away from the focus of the study and explore questions that the study did not directly address. This is especially true of lines 360-363. While the recommendations are logical, they’re not the appropriate conclusion based on the findings of the present study.

Additional comments

In the abstract, please provide the standard deviations for the swim volumes for the two groups (line 16). Also indicate the exercises that strength was tested in (line 20).

In line 101, it says “four PAP studies,” but only two of the studies discussed above were of PAP.

In all instances where 1RM strength is mentioned, please edit to "predicted 1RM strength" since 1RM was not actually tested (e.g., line 238).

In the Results (line 243), a percentage improvement in predicted 1RM strength might be a more valid comparison than raw scores, given that most people tend to hip thrust more than they squat (so we would expect a larger raw improvement). For the discussion, is it possible participants improved in the hip thrust more than the squat because they had less previous experience with the hip thrust?

In the results, please provide more detail about the moderate effect sizes (lines 246-247). How many of the 7 were SJ variables, and how many were block performance variables? How many favored the HF group and the VF group? Some of this information is contained in the discussion (lines 274-275) and would be better served in the results. The discussion would be a good place to discuss the practical significance of the magnitude of the observed changes in these metrics.

Line 279: remove “relatively”

Lines 278-286: Were there any differences between participants across the studies cited that could explain why some of them showed improvements in 6-9 week interventions while other studies didn’t?

Lines 332-336: This introduction of “individual responses to different training programs” seems to come out of left field.

---

## Round 0.2 · Minor Revisions

· Academic Editor

Minor Revisions

Thank you for your revision. The article has certainly been improved. As you can see from the reviewer comments there remains only minor amendments now. Please see their comments for specifics.

Thank you.

·

Basic reporting

No further comments.

Experimental design

No further comments.

Validity of the findings

No further comments.

Additional comments

Thank you for giving me the opportunity to review the revised version of your manuscript. And thank you for your patience.
I think that you've done a excellent job with the revisions so thank you for making my job so much easier. I'm happy with the revised version and have no more feedback to provide.

·

Basic reporting

The authors did a nice job addressing my feedback here. I noted a few instances of typos and suggested revisions to punctuation. Please see highlights/track changes in attached PDF.

Experimental design

The authors did a nice job addressing my feedback here; I have no further comments.

Validity of the findings

The authors did a nice job addressing the majority of my feedback here. However, while the authors did amend line 254, the language surrounding a "trend" for a difference in strength gains is still problematic. If there was no statistically significant difference, that should simply be stated, while also indicating the groups' scores and effect size of the difference.

Additional comments

The authors have done a nice job with their revisions. Kudos for your diligence. Please see highlights/track changes in attached PDF for my suggested revisions for typos and punctuation.

---

## Round 0.3 · accepted · Accept

· Academic Editor

Accept

Thank you for responding to the previous comments from the reviewer. I am pleased to accept this paper into PeerJ. Congratulations.